# Anti-nucleocapsid SARS-CoV-2 antibody seroprevalence in previously infected persons with immunocompromising conditions— United States, 2020–2022

Anna Bratcher[1]*, Jefferson M. Jones[2], William A. Meyer, III[3], Rehan Waheed[3], Huda Yazgi[3], Aaron Harris[4], Adi V. Gundlapalli[4], Kristie E. N. Clarke[4]

1 Epidemic Intelligence Service, US Centers for Disease Control, Atlanta, Georgia, United States of America, 2 National Center for Immunization and Respiratory Diseases, US Centers for Disease Control, Atlanta, Georgia, United States of America, 3 Quest Diagnostics, Secaucus, NJ, United States of America, 4 Office of Public Health Data, Surveillance and Technology, US Centers for Disease Control, Atlanta, Georgia, United States of America

* tqx5@cdc.gov

**Data Availability Statement:** Data procured via CDC contract #75D30122C13386 with Quest is

## Abstract

People with immunocompromising conditions (IC) are at increased risk of severe COVID-19 and death. These individuals show weaker immunogenicity following vaccination than individuals without IC, yet immunogenicity after SARS-CoV-2 infection is poorly understood. To address this gap, the presence of infection-induced antibodies in sera following a positive COVID-19 test result was compared between patients with and without IC. A commercial laboratory provided patient data gathered during July 2020–February 2022 on COVID-19 viral test results and antibody assay results, which included infection-induced (anti-N) antibody presence. Participants were categorized into having or not having IC based on if there was an indicative diagnostic code on their health record for a five-year period prior to the study period. Anti-N presence in sera from people with a positive COVID-19 test result was compared by IC status for four post-infection periods: 14–90, 91–180, 181–365, and 365+ days. A longitudinal, logistic regression produced adjusted odds ratios comparing anti-N prevalence among specimens with and without associated IC, adjusted for age, sex, residence in a metro area, and social vulnerability index (SVI) tertile. Data included 17,025 anti-N test results from 14,690 patients, 1,424 (9.7%) of which had at least one IC on record. In an adjusted comparison to patients without IC, patients with any IC were 0.61 times as likely to have infection-induced antibodies (99% CI: 0.40–0.93), during the 14–90 days following infection. Similar patterns were found when comparing people with two specific types of IC to people without any IC: (1) solid malignancies and (2) other intrinsic immune conditions. These findings stress the importance of prevention measures for people with IC, such as additional vaccination doses and consistent mask use before and after a documented infection.

governed by the following sections listed below, as written and incorporated into Modification #0002 to the contract: 1. Section C.5 – Federal Acquisition Regulation 52.227-14, Rights in Data—General, Alternate II (pg. 8); and 2. Section D.VI.B – Additional Data License Terms (pg. 26). The requested details about the data restrictions should already be listed in this FAR clause at (g)(3), as well as in the SOW section noted above. For context, it is mandatory for the federal government to implement FAR clause 52.227-14 with its Alternate II in any contract that obtains limited rights data (see requirement at FAR 27.409(b)(3)). The Limited Rights Notice language written in the FAR clause at (g)(3), as well as the language included in the SOW, came out of negotiations with and/or request by Quest. For justification, we have uploaded the contract as a document. Please find on page 11 that: "(i) The reason(s) for restricting the types of information identified in subparagraph (i) is/are: Datasets other than those available to the public as aggregate data (public use dataset available online) or approved restricted dataset with proper permissions is not allowed due to confidentiality – data that are too granular could violate the confidentiality of an individual's antibody status." Contact for data requests: Stephen Perez, MSW, PMP Product Manager, Healthcare Analytics Solutions stephen.p.perez@QuestDiagnostics.com.

**Funding:** The author(s) received no specific funding for this work.

**Competing interests:** The authors have declared that no competing interests exist.

## Introduction

While people with immunocompromising conditions (IC) are at increased risk of severe outcomes due to SARS-CoV-2 infection [1, 2], COVID-19 vaccination stimulates a weaker immune response in people with IC compared to those without IC [3, 4]. Serologically, people with IC show significantly lower seroconversion rates following primary vaccination compared with people without IC [5]. Furthermore, while people with IC who seroconvert have moderate gains in antibody titers following an additional vaccination dose, they likely remain susceptible to SARS-CoV-2 infection and accompanying severe COVID-19 outcomes [5, 6].

While the immune response following COVID-19 vaccination is well-studied in those with IC, immunogenicity following SARS-CoV-2 infection is less well understood. Since vaccination efficacy differs across individuals with and without IC, there may also be differences in post-infection immunogenicity by IC status. Furthermore, post-infection immunogenicity may also differ between IC types, similar to the variation in vaccination response across different types of IC [3]. Additionally, since IC could influence the rate at which SARS-CoV-2 antibodies are generated or wane [3], disparities may also differ by time since infection.

In this study, we seek to better understand the immune response after SARS-CoV-2 infection among people with IC to provide an enhanced evidence base for infection prevention recommendations. We compared infection-induced (anti-Nucleocapsid, designated as anti-N) antibody prevalence following a positive SARS-CoV-2 viral test result across IC status. Differences in anti-N prevalence after infection were stratified by type of IC and time since infection.

## Methods

### Study design and sample

This study examined results from anti-N SARS-CoV-2 testing performed by a single nationwide commercial laboratory network from 46 U.S. states during July 2020–February 2022. There were two inclusion criteria for this analysis: (1) a positive SARS-CoV-2 viral test result on record within the study period and (2) the presence of clinician ordered anti-N SARS-CoV-2 antibody test results following the positive viral test. Inclusion was on the patient level; data from all specimens drawn over time from an eligible patient were included in the analysis.

For each specimen, Health Insurance Portability and Accountability Act (HIPAA)-compliant records provided by the commercial laboratory included an internal unique laboratory patient identifier, SARS-CoV-2 viral test results, International Classification of Diseases-ICD10 diagnostic codes ordered by the clinician, patient age, patient sex, geographic location, and date of blood collection. Patient data on race, ethnicity, and vaccination status were not available. Data from all patients were included, regardless of the presence or absence of serological evidence of prior infection.

### Exposure

Individuals were categorized as having or not having IC based on the presence of one or more specified ICD10 diagnostic codes associated with at least one laboratory specimen drawn from the patient during the study period or over the previous five-year interval. ICs were organized into five categories: solid malignancy, hematologic malignancy, rheumatologic or inflammatory disorder, organ or stem cell transplant, or other intrinsic immune condition or immunodeficiency (S1 Table) [7].

## Outcome

Anti-nucleocapsid (anti-N) SARS-CoV-2 seroprevalence was determined with the Abbott ARCHITECT (Abbott Park, Il). This assay has an estimated 100% specificity (95% CI: 97.1–100%). Sensitivity is estimated to be 84.4% (95%CI: 66.5–94.1%) following 21 days after infection and 84.0% (95%CI: 63.1–94.7%) for 14–21 days following infection [8]. Anti-N antibodies were only produced in response to infection with SARS-CoV-2; vaccines approved or authorized in the United States at the time of data collection did not stimulate these anti-N antibodies.

**Covariates.** Data on four demographic variables were included in this analysis: (1) age group, categorized as those aged <18, 18–39, 40–64, or 65+ years, (2) sex, (3) metropolitan area, and (4) national tertile of Social Vulnerability Index (SVI). Metropolitan area and SVI tertile were determined based on the patient's county of residence. Metropolitan area was determined according to the U.S. Department of Agriculture's Rural-Urban Continuum Codes (metro 1–3, non-metro 4–9) [9]. SVI tertile was determined by ranking a patient's county of residence in the national 33.3rd and 66.7th percentiles of SVI, a county-level measure of socioeconomic and demographic factors that can affect the resilience of a community [10].

## Effect modifier: Time since infection

Anti-N test results were categorized by time since the first positive SARS-CoV-2 viral test result on record. Four post-infection time periods were examined: 14–90, 91–180, 181–365, and 365+ days following the first positive SARS-CoV-2 viral result for that patient. A sensitivity analysis compared results for the 14–90-day period to two other possible categorizations: 21–90 or 28–90 days.

## Statistical analysis

Generalized linear mixed effect models using a binomial distribution and logit link examined: (1) differences between those with any IC and those without any IC on record and (2) differences by IC type, with the comparison group being those without any IC on record. Generated estimates were adjusted for age group, sex, residence in a metropolitan area, and social vulnerability index (SVI) tertile as fixed effects. Participant was included as a random effect in all models.

We considered 99% Confidence Intervals (CI) that do not cross the null to be statistically significant and did not correct for multiple comparisons. All statistical analyses were conducted using R statistical software (version 4.1.2; The R Foundation).

## Ethics statement

This activity was reviewed by CDC, approved by respective institutional review boards, and conducted consistent with applicable federal law and CDC policy. Informed consent was not gathered, as all data were deidentified and Health Insurance Portability and Accountability Act (HIPAA)-compliant.

## Results

Of 108,323 patients who received a SARS-CoV-2 viral test prior to an anti-N test, 14,690 (13.6%) patients had a positive viral test result (Table 1). The final data set included 17,022 anti-N test results from these 14,690 patients. Patients in both the IC group and the group without IC had a median of one anti-N test on record with a range from 1–16. Of all patients with anti-N test results, 1,424 (9.7%) had at least one IC on record.

**Table 1. Sample characteristic for patients with a positive SARS-CoV-2 viral test result in the National Commercial Laboratory Study (NCLS), stratified by immunocompromising conditions (IC) status—United States, July 2020–February 2022.**

|  | No IC | | Any IC | |
|---|---|---|---|---|
|  | **n** | **%** | **n** | **%** |
| Total | 13,266 | 100 | 1424 | 100 |
| Age group |  |  |  |  |
| *<18* | 595 | 4.5 | 66 | 4.6 |
| *18–39* | 3980 | 30.0 | 193 | 13.6 |
| *40–64* | 6560 | 49.4 | 805 | 56.5 |
| *65+* | 2071 | 15.6 | 353 | 24.8 |
| *missing* | 60 | .5 | 7 | .5 |
| Sex |  |  |  |  |
| *Male* | 5140 | 38.7 | 395 | 27.7 |
| *Female* | 8099 | 61.1 | 1023 | 71.8 |
| *missing* | 27 | .2 | 6 | .4 |
| Urbanicity |  |  |  |  |
| *Non metro* | 1218 | 9.2 | 134 | 9.4 |
| *Metro* | 11972 | 90.2 | 1273 | 89.9 |
| *missing* | 76 | .6 | 17 | 1.2 |
| SVI |  |  |  |  |
| *Lowest tercile* | 2630 | 19.8 | 333 | 23.4 |
| *Middle tercile* | 3810 | 28.7 | 363 | 25.5 |
| *Highest tercile* | 6750 | 50.9 | 711 | 49.9 |
| *missing* | 76 | .6 | 17 | 1.2 |

About half of patients (49.4% of those without IC, 56.5% of those with any IC) were aged 40–64 years. Among those without IC, there were more patients aged 18–39 years (30.0%) compared to those aged 65 years or more (15.6%). This relationship was the opposite for those with any IC (24.8% aged 65 or more years compared to 13.6% aged 18–39 years. Both groups had few patients less than 18 years (4.5% and 4.6%). For those without IC, nearly two-thirds of the sample were female (62.1%), while the remaining 37.7% were male. This imbalance was more pronounced among those with any IC (71.8% female, 27.7% male). A large majority of the sample resided in a metropolitan area (90.2% without IC; 89.9% with any IC). Our sample overrepresented the highest SVI tertile (50.9% without IC, 49.9% with any IC) compared to the middle (28.7% without IC, 25.5% with any IC) and lowest (19.8% without IC, 23.4% with any IC) tertiles.

In a comparison adjusted for age group, sex, metropolitan status, and SVI tertile, sera from previously infected patients with any IC were 39% less likely to test positive for anti-N antibodies during 14–90 days following infection compared to sera from previously infected patients without any IC (aOR 0.61, 99% CI: 0.40–0.93; Table 2). Similarly, serum specimens from those with solid malignancies other intrinsic immune conditions were less likely to have infection-induced antibodies during the 14–90 days following infection than serum from those without any IC, after adjusting for age group, sex, metro status, and SVI tertile. No comparisons beyond 90 days after infection showed evidence of an association between IC and anti-N seroprevalence.

Results for the solid malignancy group were similar in the sensitivity analyses testing three possible acute post-infection periods: 14–90, 21–90, or 28–90 days (S2 Table). Results for both any IC and the other conditions categories were only significant for the 14–90-day period.

**Table 2. Adjusted odds ratios for anti-N seroprevalence in various time periods following a positive SARS-CoV-2 viral test result by immunocompromising conditions (IC)—United States, July 2020–February 2022.**

| IC status, prior 5 years | Adjusted Odds Ratios* | | | | | | | | | | | |
|---|---|---|---|---|---|---|---|---|---|---|---|---|
| | 14–90 days | | | 91–180 days | | | 181–365 days | | | 365+ days | | |
| | OR | LL | UL | OR | LL | UL | OR | LL | UL | OR | LL | UL |
| *No IC* | reference | | | reference | | | reference | | | reference | | |
| Any IC | 0.61 | 0.40 | 0.93 | 0.81 | 0.52 | 1.25 | 1.00 | 0.74 | 1.37 | 1.00 | 0.73 | 1.38 |
| *Solid malignancy* | 0.03 | 0.00 | 0.51 | 0.50 | 0.11 | 2.30 | 0.42 | 0.10 | 1.80 | 1.33 | 0.47 | 3.80 |
| ICD10 codes: C00–C80, C7A, C7B, D3A, Z51.0, Z51.1 | | | | | | | | | | | | |
| *Hematologic malignancy* | 0.42 | 0.09 | 1.98 | 0.64 | 0.11 | 3.86 | 2.36 | 0.68 | 8.13 | 0.88 | 0.29 | 2.69 |
| ICD10 codes: C81–C86, C88, C90–C96, D46, D61.0, D70.0, D61.2, D61.9, D71 | | | | | | | | | | | | |
| *Rheumatologic or inflammatory disorder* | 0.67 | 0.40 | 1.11 | 0.78 | 0.46 | 1.32 | 1.00 | 0.70 | 1.43 | 0.97 | 0.66 | 1.40 |
| ICD10 codes: D86, E85 [except E85.0], G35, J67.9, L40.54, L40.59, L93.0, L93.2, L94, M05–M08, M30, M31.3, M31.5, M32–M34, M35.3, M35.8, M35.9, M46, T78.40 | | | | | | | | | | | | |
| *Organ or stem cell transplant* | - | | | - | | | - | | | - | | |
| ICD10 codes: T86 [except T86.82–T86.84, T86.89, and T86.9], D47.Z1, Z48.2, Z94, Z98.85 | | | | | | | | | | | | |
| *Other intrinsic immune condition* | 0.52 | 0.27 | 0.97 | 0.75 | 0.36 | 1.54 | 1.01 | 0.61 | 1.67 | 1.05 | 0.64 | 1.74 |
| ICD10 codes: D27.9, D61.09, D72.89, D80, D81 [except D81.3], D82–D84, D89 [except D89.2], K70.3, K70.4, K72, K74.3–K74.6 [except K74.60 and K74.69], N04, R18 | | | | | | | | | | | | |

OR = Odds Ratio, LL = Lower limit of 99% Confidence interval, UL = Upper limit of 99% Confidence interval,— = results excluded due to imprecise estimates, shaded cells indicate statistical significance at 99% confidence

*adjusted for age, sex, metro status, and SVI tertile

## Discussion

From 14–90 days following SARS-CoV-2 infection, sera from people with any IC were less likely to contain anti-N antibodies compared to sera from those without any IC. Similar patterns were found when comparing people with two specific types of IC to people without any IC: (1) solid malignancies and (2) other intrinsic immune conditions.

Conversely, we did not find evidence of any differences in anti-N seroprevalence beyond three months following an initial positive SARS-CoV-2 viral test. A sensitivity analysis of various lags (14–, 21–, or 28–90 days) following infection supported the hypothesis that people with IC have delayed immunogenicity following infection. Another possible reason that the lack of observed differences beyond 90 days following a first infection is due to increased reinfection rates for those with IC in this period. In this case, increased reinfection could have equalized seroprevalence between IC statuses by stimulating anti-N antibody production in the later time periods at a higher rate for those with IC compared to those without IC.

These results are associated with at least four limitations. First, ICD 10 codes are not absolute indicators of IC. It is possible that patients could be misclassified by IC status in our analysis due to undiagnosed IC, omissions of an IC diagnosis made at another facility, or clerical errors in their medical record. Similarly, using a 5-year history of ICs on record may not have reflected each patient's current immune status. Second, for IC categories that specify conditions treated with immunosuppressive therapy (solid or hematologic malignancies and solid organ or stem cell transplants), ICD10 codes are a proxy for immunosuppressed status, which assume that the person is currently receiving the standard of care treatment. Third, ICD 10 codes do not indicate severity of examined conditions, either IC or COVID-19 severity. It is possible that our findings are attenuated by the inclusion of mild IC. Similarly, our observed seroprevalence differences could be produced by systematic differences in COVID-19 severity across IC categories. Severe disease may be associated with higher antibody levels [11]. Finally,

vaccination history of individuals contributing specimens were not available from commercial laboratory data, precluding stratification on this important variable.

This study highlights the disparities between infection-induced antibody seroprevalence during the 14–90 days following a SARS-CoV-2 infection when comparing people with and without IC. These findings stress the importance of infection prevention for people with IC, including consideration for additional vaccination doses, ventilation of indoor spaces, and consistent mask use with a well-fitting, high-quality mask or respirator [12]. Physicians treating individuals with ICs should maintain awareness of current availability of pre-exposure prophylaxis regimens and provide early antiviral treatment if a patient with IC becomes infected [13].

## Supporting information

**S1 Table. ICD10 codes used to categorize IC types.**
(DOCX)

**S2 Table. Adjusted odds ratios for anti-N seroprevalence in various time periods following a positive SARS-CoV-2 viral test result by immunocompromising conditions (IC)—United States, July 2020–February 2022.**
(DOCX)

## Acknowledgments

We would like to thank Elizabeth Cole-Greenblatt and Brooke Swanson for their support with this work.

## Author Contributions

**Conceptualization:** Jefferson M. Jones, Adi V. Gundlapalli, Kristie E. N. Clarke.

**Data curation:** William A. Meyer, III, Rehan Waheed, Huda Yazgi, Adi V. Gundlapalli, Kristie E. N. Clarke.

**Formal analysis:** Anna Bratcher, Kristie E. N. Clarke.

**Funding acquisition:** Adi V. Gundlapalli, Kristie E. N. Clarke.

**Investigation:** Anna Bratcher, Jefferson M. Jones, William A. Meyer, III, Aaron Harris, Adi V. Gundlapalli, Kristie E. N. Clarke.

**Methodology:** Anna Bratcher, Adi V. Gundlapalli, Kristie E. N. Clarke.

**Project administration:** William A. Meyer, III, Rehan Waheed, Huda Yazgi, Kristie E. N. Clarke.

**Resources:** Jefferson M. Jones, William A. Meyer, III, Rehan Waheed, Huda Yazgi, Kristie E. N. Clarke.

**Software:** Anna Bratcher, Kristie E. N. Clarke.

**Supervision:** Jefferson M. Jones, Aaron Harris, Adi V. Gundlapalli, Kristie E. N. Clarke.

**Validation:** Anna Bratcher, Aaron Harris, Adi V. Gundlapalli, Kristie E. N. Clarke.

**Visualization:** Kristie E. N. Clarke.

**Writing – original draft:** Anna Bratcher, Kristie E. N. Clarke.

**Writing – review & editing:** Anna Bratcher, Jefferson M. Jones, William A. Meyer, III, Rehan Waheed, Aaron Harris, Adi V. Gundlapalli, Kristie E. N. Clarke.

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
