## [Decision Letter · Decision Letter 0]

4 Sep 2024

PONE-D-24-17844Anti-Nucleocapsid SARS-CoV-2 Antibody Seroprevalence in Previously Infected Persons with Immunocompromising Conditions — United States, 2020–2022PLOS ONE

Dear Dr. Bratcher,

Thank you for submitting your manuscript to PLOS ONE. After careful consideration, we feel that it has merit but does not fully meet PLOS ONE’s publication criteria as it currently stands. Therefore, we invite you to submit a revised version of the manuscript that addresses the points raised during the review process.

We look forward to receiving your revised manuscript.

Kind regards,

Timothy J Wade, Ph.D

Academic Editor

PLOS ONE

3. For studies involving third-party data, we encourage authors to share any data specific to their analyses that they can legally distribute. PLOS recognizes, however, that authors may be using third-party data they do not have the rights to share. When third-party data cannot be publicly shared, authors must provide all information necessary for interested researchers to apply to gain access to the data. (https://journals.plos.org/plosone/s/data-availability#loc-acceptable-data-access-restrictions)

Reviewers' comments:

Reviewer's Responses to Questions

**Comments to the Author**

1. Is the manuscript technically sound, and do the data support the conclusions?

Reviewer #1: Yes

Reviewer #2: Partly

2. Has the statistical analysis been performed appropriately and rigorously? 

Reviewer #1: Yes

Reviewer #2: No

3. Have the authors made all data underlying the findings in their manuscript fully available?

Reviewer #1: No

Reviewer #2: No

4. Is the manuscript presented in an intelligible fashion and written in standard English?

Reviewer #1: Yes

Reviewer #2: Yes

5. Review Comments to the Author

Reviewer #1: Dear Editor,

The authors have done a nice job in putting up the manuscript and presenting it in a concise and clear manner. The topic of immune responses to a respiratory infection in the immunocompromised is apt and the findings may offer insight for public health approach to such infections.

Having said that the authors can respond and clarify a few observations in the manuscript.

1. The nucleocapsid antibodies were assayed using two kits from different manufactures (Abbott and Roche). Commercial assays used to detect anti-N antibodies have highly variable performances with significant loss of sensitivity with time postinfection (the performance specifications of the kits are not provided). However, the authors have not shared information on the distribution of the two assays between the immunocompromised and the non-immunocompromised- and whether this influenced the study findings.

2. The vaccination status of participants is unknown. A study reported that anti-N antibody responses were lower in plasma after SARS-CoV-2 infection in vaccinated patients compared with patients infected before vaccination or infected without vaccination. Again, since information is not provided one cannot tell what impact this would have had on the study findings especially if those who were IC had been encouraged to get the shot and ended with higher proportion of vaccinated compared with non-IC.

3. Severity of SARS-CoV-2 infections is not provided, and this would have had an impact on the response and duration of anti-N antibodies. Did the IC have more severe infections than the non-IC or the vice versa?

Reviewer #2: In the manuscript “Anti-Nucleocapsid SARS-CoV-2 Antibody Seroprevalence in Previously Infected Persons with Immunocompromising Conditions — United States, 2020–2022” by Bratcher et al, the authors analyze data from Quest looking at the ability of immunocompromised and non-immunocompromised patient’s ability to mount an infection-induced antibody response to SARS-CoV-2 infection. Overall I think this paper was well researched but I believe there are a few areas that need improvement before publication. See specific questions and notes below:

Methods:

-Authors should define ICD10 codes used to categorize patients. This could be a supplemental figure, but each code used should be defined. “Other intrinsic immune conditions” should also be defined.

-Methodology for serology was defined but specific viral test was not. Please include.

-Statistics – it was mentioned that statistics were not corrected for multiple comparisons, but I they should have been. I would consult with a statistician to be sure. Also, you mention p values, but none are presented. Please include them.

-Were previously infected patients (those with positive serology before viral testing) included in analysis or were they left out?

-Could you better define sensitivity analysis and what you compared?

Results:

-It would be helpful if you include a demographics table that included the typical age, sex, SVI, metropolitan data, how many patients were previously infected before viral test, etc for both non and IC groups.

-Table 1: If using “LL” and “UL” you must define it under the table. You could also consider turning these numbers into a graphic by graphing the range and noting the OR with a symbol. Graphing in this way and including a line at 1 could help distinguish the significant findings. I assume the blue highlight is to also distinguish the significant values, but that should be defined somewhere if you decide to stick to this table format.

-Sensitivity analysis is mentioned below the table as being similar but you need to define that data.

Discussion:

-You mention you looked at sensitivity analysis of other time periods (14, 21, 28-90)[lines 129-130] but this wasn’t in the methods or results. Please provide if you are going to include it in the discussion.

-You mention increased reinfection rates [line 132] in the discussion but don’t provide that data. You could include this in the demographics table I mention above.

-Limitations: It would be good to mention that the time frame of looking at ICD10 codes 5 years prior may not accurately represent the patient’s immune status at time of testing.

6. PLOS authors have the option to publish the peer review history of their article (what does this mean?). If published, this will include your full peer review and any attached files.

Reviewer #1: **Yes: **Daniel Maina

Reviewer #2: No

---

## [Author Response · Author response to Decision Letter 0]

18 Oct 2024

Thank you for all the thoughtful comments and edit suggestions. Please find our responses in the included "Plos one Response to Reviewers" document.

---

## [Editor Report · Decision Letter 1]

29 Oct 2024

Anti-Nucleocapsid SARS-CoV-2 Antibody Seroprevalence in Previously Infected Persons with Immunocompromising Conditions — United States, 2020–2022

PONE-D-24-17844R1

Dear Dr. Bratcher,

We’re pleased to inform you that your manuscript has been judged scientifically suitable for publication and will be formally accepted for publication once it meets all outstanding technical requirements.

Kind regards,

Timothy J Wade, Ph.D

Academic Editor

PLOS ONE
---

## [Editor Report · Acceptance letter]

5 Nov 2024

PONE-D-24-17844R1 

PLOS ONE

Dear Dr. Bratcher, 

I'm pleased to inform you that your manuscript has been deemed suitable for publication in PLOS ONE. Congratulations! Your manuscript is now being handed over to our production team.

Kind regards, 

on behalf of

Dr. Timothy J Wade 

Academic Editor

PLOS ONE